# Designing Mobile Health Applications to Support Walking for Older Adults

**DOI:** 10.3390/ijerph20043611

**Published:** 2023-02-17

**Authors:** Yasmin Felberbaum, Joel Lanir, Patrice L. Weiss

**Affiliations:** 1Information Systems Department, University of Haifa, Mt. Carmel, Haifa 3498838, Israel; 2The Helmsley Pediatric & Adolescent Rehabilitation Research Center, ALYN Hospital, Jerusalem 9109002, Israel

**Keywords:** physical activity, older adults, field study, mobile health applications

## Abstract

Physical activity is extremely important at an older age and has major benefits. There is a range of applications that help maintain physical activity. However, their adoption among older adults is still limited. The purpose of the study is to explore the key aspects of the design of mobile applications that support walking for older adults. We conducted a field study with older adults, aged 69–79 years, using a technology probe (a mobile application developed as an early prototype) with the purpose of eliciting requirements for mobile health applications. We interviewed the participants during and after the study period, asking them about their motivation for walking, usage of the application, and overall preferences when using such technologies. The findings suggest that mobile applications that support walking should address a range of walking variables, support a long-term learning process, and enable the user to take control and responsibility for the walk. In addition, we provide design guidelines concerning the motivation for walking and the data visualization that would make technology adoption easier. The findings from this study can be used to inform the design of more usable products for older users.

## 1. Introduction

Physical activity is extremely important at an older age and has major benefits including maintaining flexibility, balance, normal blood pressure, and fitness [1]. It also reduces psychological difficulties such as the fear of moving outdoors [2] or the fear of falling [3]. General quality of life is also highly affected by performing physical activity [1]. According to the World Health Organization, walking at an older age helps reduce heart disease, osteoporosis, and diabetes among other age-related issues and contributes to interpersonal communication and social interaction. In addition, it is an activity affordable for all people, enjoyable to most, and thus, a potentially feasible way to maintain active healthy aging [4].

There are various applications that help maintain a long-term physical activity routine based on monitoring performance. For example, Fitbit™ (Alphabet Inc., US) (https://www.fitbit.com/global/us/home) and other smartwatch companies offer different activity tracking products and services to better support physical activity (e.g., full workout plans); RunKeeper™ (ASICS, US) (https://runkeeper.com/cms/) and Strava™ (Strava Inc., US) (https://www.strava.com/about) are activity tracking mobile applications for runners and cyclists. Similarly to Fitbit™ (Alphabet Inc., US), these applications provide basic tracking and insights free of charge, which can be upgraded by paying a monthly/yearly fee. These applications were shown to be significant tools for creating habits and supporting behavioral change [5,6]. However, despite the existing range of products and their benefits, technology adoption with these tools is still low among older adults compared to younger adults [7,8]. This may be due to difficulty in understanding a device’s functional features or not being convinced of the added value of such technologies [9].

The purpose of this paper is to present key recommendations for designing walking mobile applications for older populations that would be better used and adopted. Next, we use the Personal Investment Theory (PIT), which explores how to engage people in physical activity, to describe the older adult walker, including personal attributes and personal incentives related to the walking activity, walking physical activity at an older age [10,11], and current technologies and applications targeted at walking for older adults.

### 1.1. The Older Adult Walker

This section describes the main characteristics of older adults that may affect walking. These are divided into personal and physical attributes, and incentives and motivations for walking as presented in Figure 1.

#### 1.1.1. Personal and Physical Attributes

Older adults experience a well-documented decline in physical ability, including decreases in muscle strength and power, balance, mobility, and joint flexibility [1]. In addition, chronic pathologies such as those that affect the respiratory system (e.g., chronic obstructive pulmonary disease), the cardiovascular system (e.g., pulmonary hypertension or heart disease), and the musculoskeletal system (e.g., stroke) affect physical ability as well. In addition, as people age, they experience a decrease in cognitive capabilities, such as memory decline expressed as a decrease in working memory (the ability to keep and manipulate information), semantic memory (acquiring knowledge), or procedural memory (the steps needed to do a specific task); age-related attention issues, such as a decrease in selective attention (searching for a specific object), which leads to a need for cues to orient and to attract attention (e.g., on a screen while using an application); or difficulty to perform dual-tasking (walking while doing another activity) [12,13]. There is also evidence for a decline in spatial cognition [12] and spatial navigation [13]. Being physically active has a major impact on physical condition (functional capacity, decreased risk of chronic disease), quality of life, and life expectancy [1]. It leads to a healthier lifestyle, social networking, and a positive mental attitude and is associated with significant improvements in overall psychological health [1].

PIT relates to a person’s sense of self, which includes self-perception, competence, and beliefs towards performing physical activity [14]. Practically, it refers to having the control and responsibility to maintain a healthy lifestyle [14]. Older people tend to compare their competence to younger relatives and acquaintances [15] and, most often, to their younger selves [16], which can lead them to underestimate their current capabilities [14]. Some of the abilities described above may change over time, even on a daily basis. For example, occasionally, a person may not feel well or be in the mood for exercise. Therefore, it is important to recognize both the general sense of self and current state of a person when considering encouragement of physical activity.

#### 1.1.2. Personal Incentives and Motivation of Older Adults to Perform Physical Activity

Motivation can be categorized into two types: intrinsic and extrinsic. Intrinsic motivation relates to doing an activity for internal reasons, such as enjoyment or satisfaction gained from the activity. Extrinsic motivation is an external reason for doing an activity, such as recognition, peer pressure, reward, or incentives [17]. When focusing on physical activity, improving existing skills or acquiring new ones may be strong intrinsic motivators to perform more activity [18]. In addition, external social motivations such as competition, social comparison, or cooperation towards a mutual goal can enrich and enhance the activity [19].

Motivation for older adults differs somewhat. In general, social recognition and competition are less important for older adults [14]. With regard to walking, motivators are divided into intrapersonal and interpersonal [20,21]. Intrapersonal motivators include intrinsic motivators such as the will to maintain or improve one’s physical condition (e.g., balance, walking ability, and strength), mental/emotional goals (e.g., stress relief, self-image, reducing depression), and personal progress towards one’s goals. Interpersonal motivators relate to social aspects (both intrinsic and extrinsic) of physical activities, such as opportunities to interact and communicate with friends and peer support (being part of a group) [14,20,21].

### 1.2. Walking Activity among Older Adults

We present the literature on the properties of a walk relevant to older adults using the third PIT component, perceived options and opportunities to perform physical activity [14], which is summarized in Figure 2.

Perceived options for physical activity are evaluated by the walker according to three factors [14]. First are the perceived consequences of the activity. Walking is known to provide benefits (e.g., improve one’s mood, strength, and balance) or to have some risks (e.g., knee strain, falling). Older adults are more sensitive to these consequences than younger adults (e.g., slower gait, risk, and fear of falls) and therefore they evaluate the consequences of an activity more carefully [3]. Second, older adults evaluate barriers and facilitators of an activity to make decisions about the walk. Previous work identifies challenging physical environmental features, such as inclines, obstacles, and weather versus attractive features such as areas with less pollution and more greenery. The availability of benches, parks, and places to buy food and drinks are also very important facilitators [21,22,23]. A third factor is an understanding of the walker regarding how the walk enhances personal goals. For instance, if a 30-min walk helps regulate blood sugar, the goal is very clear. Common ways to measure walking goals include steps, time, distance, etc.

#### Current Research, Technologies, and Applications That Support and Enhance Walking Activity among Older Adults

Many technologies and mobile applications aim to enhance and maintain physical activity. Most smartwatches provide the option of tracking various activities (e.g., walking, running, swimming, strength exercise) using smartwatch sensors and a mobile application for adjusting settings and displaying the activity performance. Other mobile-only applications may monitor walking and running activities and present the user’s performance and route. Extensive research has explored different aspects of applications that encourage physical activity. Rabin et al. [21] asked participants to compare three commercial applications supporting physical activity after using them for one week. The participants discussed existing features such as automatic tracking supporting various physical activities as well as new features, such as integrated music and uploading a daily photo. Traunmueller et al. [22] explored goal-oriented walking versus experience-oriented walking with the Space Recommender System, a wayfinding planner that focuses on the quality and enjoyment of a walk (e.g., walking through preferred points of interest). Ding et al. [23] focused on reminders as a way of promoting walking and showed that context-based reminders are better than random reminders for this purpose.

The above-mentioned studies all explored walking applications designed for young healthy adults. However, when dealing with walking, it is clear that the ability and needs of older adults are very different. Only a few studies focused on exploring walking applications for older adults. Albaina et al. [24] presented a design process in which a stationary walking display was described, using a flower metaphor to represent users’ performance (in walking steps) [24]. Helbostad et al. [25] describe the development process of PreventIT, a physical activity mobile application and smartwatch for tailored exercises for the older population. Mastropietro et al. [26] present the development process of NESTORE, a platform for e-coaching for healthy aging. Yerrakalva et al. [27] and Solis-Navarro et al. [28] indicated the importance of applications that enhance and support physical activity to improve function among older adults.

As with younger adults, mobile technologies have the potential to support and enhance physical activity for older adults. However, despite the variety of products and the benefit the older population may have from mobile health technologies, their adoption rate has been relatively low, and over 43% of people aged 70 years old and above quit using mobile health technology after only 14 days [29]. According to Tajudeen et al. [30], this happens due to complicated features and menus as well as interface designs that do not take into consideration the older population. Therefore, the goal of the present paper is to determine how to design mobile health technologies for walking targeted at supporting the older adult walker.

The purpose of the current work is to improve the design of mobile applications that will better support the physical activity of walking among older adults. To that end, we designed and built a mobile application, *MyWalk*, to serve as a technology probe that would help us better understand older adults’ needs from such applications. A technology probe is an early version of a product intended to be used by potential users in their natural environment during the early stages of the design process [14,31]. Therefore, our intention in *MyWalk* was not to introduce a new application, but rather to learn about the use of similar products and elicit design insights for future ones. *MyWalk* provides recommendations and monitoring of convenient walking routes near one’s home, as well as several real-time support options designed specifically for the older population. We conducted a 2–3-week field study of older adults aged 69–79 years. The collected data as well as the subjective feedback from the participants served as a springboard for discussion on current and future mobile applications for walking. We present the findings from the study as well as our key recommendations for designing mobile walking applications for the older population.

## 2. Methodology

In this section, we describe the methodology of the study including a description of the application we developed as a technology probe.

### 2.1. Participants

Fourteen (14) participants took part in the field study (3 females, 11 males). The participants were recruited through social media (mainly through Facebook and neighborhood WhatsApp groups) as well as through community centers. The inclusion criteria included participants who could walk at least 30 min twice a week, used a smartphone on a regular basis, and owned an Android-based mobile device. Participants’ ages ranged from 69–79 years (*M* = 72.6, *SD* = 3.6). All the participants are retired, although two still work part-time. The distance the participants typically walk ranges from 1.5 to 10 km (*M* = 4.9, *SD* = 3.1). Some of the participants already use mobile health applications such as LG Health™ (LG H&H, South Korea) (https://play.google.com/store/apps/details?id=com.lge.lifetracker&hl=en&gl=US, accessed on 30 September 2022) and pedometers for walking. Others use location-based applications such as Google maps™ (Google Inc., USA) or Altimeter. Informed consent was obtained from all subjects involved in the study. Specific details about the participants are presented in Table 1.

### 2.2. Physical Activity Status (PASE) Questionnaire

To assess participants’ physical activity levels, we administered the PASE questionnaire [32]. Scores can range from 0 to 400 [32], and the norms reported according to the PASE manual [32] for the 70–75 year age group are *M* = 102.4, *SD* = 53.7 for men and *M* = 89.1, *SD* = 55.5 for women. In the current study, scores ranged between 33.6 and 131.8 (*M* = 78.1, *SD* = 32.6). Considering that the majority of our participants are males, their general physical activity level is relatively low.

### 2.3. MyWalk Technology Probe

We conducted a 2–3-week field study using a customized technology design probe application. A design technology probe (or rapid technology probe) is a working system that can be presented to users at the early stages of a design process to receive initial feedback and elicit ideas and requirements for the final product [33]. Its goal is to collect information about the use and the users of the technology in a real-world setting, to field-test technology, and most importantly to inspire designers to think of how technology can support users’ needs and desires [14,33,34]. It is often used when addressing a population with special needs such as older adults [31,32]. Thus, rather than evaluating the application per se or creating a behavioral change, our main goal was to identify requirements for future walking mobile applications so that they will be adopted and used more consistently by older adults. As explained by Hutchinson et al. [35], technology probes are helpful in two ways: (1) they enable data collection in the real daily users’ environments (e.g., field studies [36]); and (2) they address both social and technological issues, enabling users to reflect on the use of the technology in various ways in a specific situation (e.g., by documenting their experiences in diaries). Unlike a final product, technology probes provide users with opportunities to interpret and elicit their own ideas during the design process [34], even in the early stages [33]. In this study, we developed a mobile application design probe for eliciting requirements and extracting the main aspects older adults need or would like to have in a walking support mobile application. Developing a probe, unlike a commercial product, enabled us to test new features (e.g., walking routes), adapt the application to older adults, and collect log data.

#### 2.3.1. Design Process

We designed and developed a mobile application called *MyWalk* to serve as our technology probe. Previous research on physical activity mobile applications for older adults indicated the significance of setting goals and monitoring activity in walking as well as the need for route planning options, such as the option to learn new routes and make known routes more interesting or challenging [37]. Therefore, *MyWalk* enables its users to monitor their walks and performance similar to other physical activity applications. It also has a new feature of suggesting circular walking routes around one’s home location.

Throughout the design process, we followed user interface recommendations for the design of mobile applications for older adults : high contrast between text and background with the suggested route drawn on the map in red, and user’s actual walking in real time drawn in blue; preference of text labels over icons (icons should be familiar in order to be understandable); using the participants’ native language in all the application components (including the map); and using a 12-point text font size where the main information is textual (e.g., walking settings) [12].

As suggested in previous work, we provided users with full control over their physical activity via customization [38]. This is expressed in the following ways: (1) suggesting routes to the user to choose from; (2) allowing the user to choose before each walk whether they would like to walk more, less, or the same distance as the previous walk; and (3) the possibility of changing their goals and walking settings any time (i.e., weekly walking goal distance, suggested routes distance range, and the minimal number of sitting points).

With regard to visualization of performance, previous research indicated that older adults find added value in general overviews rather than specific information, where they can gain insight from the data [39]. Therefore, simplifying data visualization by presenting the minimum data needed and limiting a graph/chart to up to three variables is recommended [40]. Moreover, the older population was shown to prefer a bar chart over other chart styles, reporting that it is “familiar, provides quick longitudinal comparisons” [41]. Thus, we use a simple bar chart to present the distance and duration of walking over three granularities of performance (daily, weekly, and monthly) to facilitate users to explore their performance in detail as well as in a high-level manner.

The application was developed in Java on Android Studio platform 2.3.1 using Mapbox API and Open Street Maps data (OSM). Figure 3 presents the application’s main screens.

#### 2.3.2. Flow of Use

The users start by defining a weekly distance goal they would like to pursue (in kilometers) and choose a contact person to reach out to in case of distress during the walk. Figure 3a presents the main page. The user can see a weekly performance progress bar and information about the last walk and has the option of starting a new walk. When the user starts a new walk, the application suggests three circular routes and the option of a “free walk” (in which case, the user decides on the route). Since this is a probe, we made a broad assumption that people usually start their walk from their home and therefore, the participants could receive circular route suggestions starting from their home only and not from any other point they were at. These suggestions are based on the user’s preferences (distance range and number of resting points are defined in the “Route settings” tab), or on their previous walk (the system asks users whether they would like to walk more/less/about the same distance as the previous walk). For each route suggestion, the user can see the distance of the route, the streets it passes through, the number of resting points, restrooms, places to eat and drink, and the elevation of the route (whether it is flat, steep, or very steep) (Figure 3b). After choosing a route, a preview of the route appears on the map.

While walking, time, distance, and user progress are monitored and the route is drawn on the map (Figure 3c). The application provides several options to support the user along the way: (1) Bench button—directs the user to the nearest sitting location; (2) SOS button—calls the predefined contact person and transmits the current address of the user; and (3) Home button that ends the walking session and directs the user back home on the shortest route. There is no need to hold or look at the device while walking unless the user needs or wants to operate the described options. After the walk, on a separate tab, users can see their performance in terms of distance and duration of walks daily, weekly, and monthly (Figure 3d). The interaction with the application is documented on the phone itself.

### 2.4. Study Procedure

We met with the participants at the beginning and end of the study. During the first meeting, participants signed an informed consent form and then received a general explanation of the goals of the study. Next, participants answered several demographic questions (e.g., age, family status, residence, and professional status). The participants then completed the PASE [32] to assess their initial physical activity status. We then installed the application on the participants’ mobile phones, provided a detailed explanation about its functionality, and let them try the application and see whether they had any further questions. Participants were asked to use the application for a 2–3-week period. Each participant was asked to walk a minimum of two times a week, 30 min each time, but of course could walk more if they wanted to. Participants were also asked to document their walks in a diary.

After one week, we initiated a short phone call in which we asked several questions regarding the participant’s walking experience with the application, the usability of the application, and the visualization of the data. This was also an opportunity to address any difficulties or misunderstandings they had with the use of the application. At the end of the study, we met with each participant again and conducted a semi-structured interview discussing the use of the application and the general perceptions of participants concerning such an application (see next Section). Then, the participants completed the System Usability Scale (SUS) questionnaire to evaluate the usability of the application [42]. We also gathered the data (log files) recorded on the participants’ mobile phones while using the application and the diaries. All interviews as well as the phone calls were audio-recorded. We provided technical help to the participants whenever it was needed by phone or text messages.

### 2.5. Outcome Measurements

In order to collect comprehensive data on the participants’ behaviors and views, we triangulated information from several input sources. Our main data sources were the semi-structured interviews and the participant’s diaries. The interview questions included three aspects: (1) walking experience with the application: we asked the participants to describe their general experience; whether they think their performance improved, worsened, or remained the same; whether they used the route suggestions and if so, whether they discovered new places in their neighborhood; and what features they would have added or removed to create a better walking experience; (2) the usability of the application. We asked about preferred and missing features; features that can be added to enhance motivation to walk and trust in the application; (3) visualization of the data. We asked participants whether and how they used the visualizations of the walking information; whether they used the daily or weekly view; whether they preferred to use the distance or duration view; and why they viewed the walking data.

In the diary logs of the walks, participants were asked to refer to the following categories: general walking experience, better and worse experiences while walking with the application, challenges related to the walk and the application, and any additional comments. The interview and diary questions were kept as open as possible to let the participants lead the conversation and add their own comments. In addition, all interactions with the application were recorded, stored locally, and gathered at the end of the study.

### 2.6. Data Analysis

Results from the PASE and SUS questionnaires were calculated according to their manuals [41,43]. Our focus was to code behaviors and identify themes that are relevant to all aspects of walking while using such a walking application. Text from all interviews and diaries were conducted in the participant’s native language and were transcribed and analyzed. We applied bottom-up, thematic analysis [44] using Atlas software (https://atlasti.com/, version 22, accessed on 30 September 2022) for qualitative analysis to create initial code groups that were later categorized into themes. Quotes from participants presented in this paper have been translated into English.

## 3. Findings

Our analysis revealed important themes related to the design of walking applications for the older population. We discuss each of these themes, describing how they emerged in the study, and tying them to previous work. Table 2 presents a summary of walking with the application during the study.

### 3.1. Supporting Various Types of Walks

Previous work differentiated types of walks according to quantitative characteristics (e.g., gait patterns [45]) or environmental settings (e.g., pedestrian density [46]). In addition, most current physical activity mobile applications provide the option of setting quantitative goals (e.g., steps, distance) and monitoring one’s performance accordingly. However, at an older age, physical activity is not always the goal of a walk. For example, a walk can serve as a way to spend time with a friend, a way to go out, or as a leisure activity.

From the interview analysis, we identified several types of walks that may affect the attributes and goals of a walk:Planned walk (setting a specific time for a walk) vs. a spontaneous walk:Walking alone vs. together (e.g., with spouse or friends). P10, for example, said: “I have a friend that likes to walk fast, and when I walk with my wife, we walk slowly, with ease”;Known route vs. unknown route;Known vs. unknown environment (e.g., when traveling);Indoors (e.g., walking on a treadmill) vs. outdoors (e.g., when walking outdoors, the walker may encounter various obstacles);Casual walk (e.g., while performing errands which may include a lot of stops) vs. a walking session;Mindful walk (e.g., enjoying the activity itself, feeling one’s muscles, breathing fresh air) [22] vs. performance-oriented walk (e.g., walking to achieve goals).

Since any walking activity is important at an older age, helping people understand and recognize their preferred type of walk at a specific time or in a specific context will lead to greater engagement and enrich one’s positive, perceived consequences of walking (in accordance with Figure 2). This is expected to promote walking as a regular activity.

### 3.2. Providing Appropriate Motivation for the Elderly

Several themes raised in the interviews related to both extrinsic and intrinsic motivation. An important extrinsic motivation factor for walking was the participants’ medical condition [47]. Two of the participants referred to their medical condition as a motive for walking due to their doctor’s recommendation. Several participants said they miss having any immediate benefit from walking although they know it is good for them in order to maintain a healthy lifestyle. They mentioned that knowing how much energy/calories they invested in their activity might help motivate them by providing an immediate reward for the activity. P3 said: “*I don’t understand what I get from these (walks)… if I could see that I burned 200 calories I could have known that I could eat another slice of bread or that I could lose 50 g*“. We note that there already exists a measure called the metabolic equivalent of task (MET), which reflects the metabolism during exercise and the intensity of the activity [47,48] in several applications (e.g., Fitbit™, MetTrackerApp™ (MetTracker, Germany) (https://www.mettracker.com/en/)). Several participants pointed out the visualization of their walk as a motivating factor. They looked at their previous walks and wanted to reproduce their successful performance so it would be reflected in the data visualization. For example, P3 shared her pleasure at seeing a tall bar chart, showing a long distance/time of walking. She shared that she was very proud of herself in these cases, which helped motivate future walks.

Intrinsic motivation refers to a willingness to improve or maintain physical condition for one’s self and enjoying the activity itself (interpersonal) as well as being social (intrapersonal). P8 shared: “*The walk is enjoyable for me regardless; I just enjoy walking*”. He also added: “*I take my children and grandchildren to walk in a nature trail, it’s just fantastic*”. In accordance with previous work concerning older adults [14], all the participants mentioned that they do not wish to compete or be compared to other people, and the only comparison is against themselves (P10: “*I would have liked to see a table with my performance. Similar to a leader board, but with my performance only*”).

### 3.3. Supporting Easy Learning and Adoption Experience

We designed the application to be as simple as possible and meet design recommendations for older adults [12], as explained in Section 2.3.1. In addition, the participants received an extensive explanation about the application during the first session; we showed participants the features of the application, described the flow of use, and let them try the system for as long as they needed to see whether they had any further questions. All participants expressed their need to dedicate additional time and attention to learning how to use the application. For some participants, it took a few hours (P5: “*When I got used to the application, I had no further problems… The eye and the finger go to the right places and everything looks easy and simple…It took me 24 h to get used to the application.*”). Others performed a few walks while trying the application until they felt comfortable with it (P10: “*At the beginning, I did short walks just to understand how it works and if it doesn’t interfere with other functions of the mobile phone, like answering the phone*”), while others chose to focus on learning the main features first and gradually got to know more features (P2 after one week into the study: “*I don’t have any opinion about the application nor have I looked on my performance yet. I take it little by little*”). Two of the participants raised the need for a printed instruction guide to be received together with the application. Indeed, P3 wrote her own instructions accompanied by sketches while we presented the application to her at our first meeting. She said: “*When you buy an iron, you receive an instruction book. It is important for me to have a flowchart of all the possible options and cases of the product*”. The SUS results were relatively high (*M =* 70.35, *SD* = 15.75), indicating that despite the longer learning process, participants perceived the system as usable.

When discussing the adoption of mobile health applications, Nunes et al. [49] referred to perceived usefulness as performance expectancy, ease of use as effort expectancy, and added facilitating conditions, meaning the belief technical support will be available, as another key factor in the adoption of mobile health technologies by older adults. Technology learning, acceptance, and adoption is a well-known challenge for the older population due to reasons such as incompatible interfaces, interaction (e.g., using precise gestures or a lot of typing), and flow of use (e.g., not being used to this technology or understanding the steps needed to operate the system) [49,50]. Specifically, perceiving a form of technology as useful and easy to use [51] and its added value [9] makes it more acceptable to older users.

### 3.4. Evaluating Performance Using Data Visualization

Previous work emphasized the importance of simplicity when the older population uses visualizations of quantitative data [40,51]. We designed the visualization in accordance with these guidelines, presenting only the most important aspects of the walking activities in a simple way using the preferred data representation for older adults, i.e., a simple bar chart [41]. Our findings indicate that this was understandable and liked by most of the participants, providing further support for these recommendations. Most of the participants (12 out of 14) viewed the walking data visualization after each walk, using both kilometer and time representations. Their reasons for looking at the data were usually to assess and self-reflect on it, e.g., comparing their distance/duration to the day before. Another reason was to make sure that their data were saved (P1:” *If I say that I walked 10 km, I need to have a proof of that*”). Two of the participants referred their performance to an experience (e.g., referring their long walks with a trip they did that week). More than half of the participants (8 out of 14) also shared their performance with others by showing the visualization to their spouse, close friends, and children in order to receive recognition. However, most of the participants commented that they would not wish to share this information on social media, but rather, preferred to physically show the visualizations on the mobile device’s screen.

### 3.5. Personalizing the Activity: Adaptation vs. Customization

Personalization is the optimization of a system’s interface according to user preferences and needs [52]. When personalization is initiated by the system, it is referred to as adaptation and when initiated by the user, as customization [53]. Both adaptation and customization are extremely important in the context of physical activity for older adults since they enable adjustment of the exercise to suit the user’s needs and abilities, which typically leads to better engagement and enjoyment [54]. Customization, specifically, is referred to as an intrinsic motivation tool providing older adults with control and autonomy over their physical routines [38] and provides the user with the opportunity to take responsibility for the exercise [14]. In our application, we provided users with control over their physical activity via several ways of customization (see Section 2.3.1). However, only 4 out of 12 participants walked using the suggested route feature. All four indicated they had explored the feature briefly and decided to give it up after a short time. The other participants preferred to do free walks and not commit to a pre-defined route suggestion. Seven participants said that although they explored the route suggestion features, they chose their own walking route. Regarding the walking settings, none of the participants changed the weekly goal we set at the first meeting. Six of the participants changed their selected resting point preferences and six participants changed their walking distance range preferences. An interesting suggestion was to provide the option to set a specific altitude of the route (P7: “*A challenging incline for one person may be easy for me*”). Various reasons were given for not using the system’s suggested routes. These reasons are detailed and further discussed in Section 4.2.

### 3.6. Providing Support during the Walk

Most of the participants enjoyed seeing their walking trajectory on the map in real time. One of the participants said: “The application is great, it shows me right where I am, not only at the end of the walk, but every time I look at the map. I can also see where I started from and where I’m heading to” (PA9). However, seeing the progress drawn as a line was not always optimal. When walking back and forth in the same area, the lines overlapped, and the walking route became less clear. An interesting topic raised was that the drawn line did not represent the level of difficulty in real-time (e.g., when going up a slope or when walking on an unpaved path). For example, P8 commented: “When I walk in nature, the marked trail and the path are unpaved and full of rocks, you need to walk carefully and slowly. […] There is no expression of level of difficulty of the way to explain the shorter walk”.

The support options during the walk (i.e., rest, SOS, and home buttons) were not used by our participants. Most of the participants indicated that they walk without resting and therefore did not need the “Bench” button. The “Home” button was not intuitive to most participants; however, after understanding its purpose, participants mentioned that they did not need this option since they were very familiar with their walking environment. The SOS button was welcomed; however, all the participants except for one did not see themselves as people who might need it (even participants who had various heart conditions or an injury). Some of the participants said they are not “the usual type of older people”, meaning they are very active.

Participants emphasized the importance of measuring physiological signals, which was missing in the probe. For example, pulse and blood pressure were suggested as measures that should be visible while walking. Furthermore, having the option of setting minimal and maximal values of physiological measures and receiving a notification when the outcomes are out of range were also mentioned as a wanted safety option. In addition, the amount of energy invested during the walk was also suggested to be presented while walking (see Section 4.2). With regard to the walk, presenting the mean walking velocity and step count was suggested by six participants. Data about the walking environment was also raised by the participants as something that they would like to get information about. Air quality and green places were suggested by four participants to be taken into account in route suggestions, as mentioned in previous work [22], in order to avoid polluted and crowded routes. Altitude data were suggested to be shown both while walking and in the data visualization as an indication of the level of difficulty of the walk. Other suggestions were related to documenting the subjective experience of the walk (in addition to quantitative measures). One of the participants suggested pinning pictures along the route (similar to Google maps options). Documenting the subjective level of difficulty was also suggested. Documenting the walks as a subjective experience also enables sharing various experiences with family and friends: “*I can show my grandchildren a beautiful cave I saw or a nice scene I ran into while walking as well as my total performance, and interest them with my story*” (P7). In addition, two participants would have liked the option of saving their favorite routes.

## 4. Discussion and Design Recommendations

In this section, we discuss the main findings raised in the previous section together with recommendations for future mobile applications to better serve the older population.

### 4.1. Walking as a Multi-Purpose Activity

In current mobile health applications, physical activity is measured mainly by quantitative measures (e.g., steps, distance, duration) [28,29,54]. Similarly, in our probe, we presented users with the distance and duration of the walks. However, we observed that other parameters affect the walk, especially at an older age, which are not considered when evaluating user performance. First, the participants indicated that they performed various types of walks based on their physical state, mood, and the people they were with at that moment. They explained that these factors can affect their measurable performance and can lead to trade-offs. For example, doing a short walk despite feeling that one is less physically well requires more effort compared to when one is feeling better and should be acknowledged. Previous research has shown that social and cognitive factors together with physical and physiological abilities, as presented in Figure 1, and a person’s sense of competence affect the physical activity level and what they are able or feel able to do [14]. Therefore, systems should strive to evaluate and interpret the user’s current physical and mental status when building a complete user model. An evaluation of the user’s state may be achieved via relevant questionnaires [53,55] and calculated through the mobile application or by using additional sensors. Further research is needed to understand how to collect these data without spoiling the user experience.

Second, in some types of walks, quantitative measures do not appear to tell the whole story. For example, walking together with a friend may result in a reduced number of steps or shorter distance; however, it may be the reason a person decides to go for a walk due to a social benefit for well-being. Walking in an unknown route or environment may result in an increased number of steps and distance due to exploration or disorientation, but with a decreased speed of walking. A mindful walk may also be slower, or less strenuous, yet being mindful and present contributes significantly to one’s mental health and is of great value [56]. Therefore, we recommend enabling documentation of the type of walk a user is taking. For example, whether the walk is done alone or with someone; in a known/unknown route and environment; and whether it is mindful or goal-oriented. In this way, users could be provided with a variety of goals and perceived consequences that can be achieved by walking (see Figure 2).

Third, the personal facilitators and barriers of a walk (Figure 2) have a large impact on the walking experience and level of difficulty for older adults and should also be considered when measuring and reflecting on the walk. Participants expressed the need to see an indicator of their level of difficulty, both while walking (e.g., by changing the color of the drawn walking route) and in the data visualization, in order to have an explanation in case of “poor performance”.

### 4.2. Control and Responsibility towards the Activity

The need for privacy and a sense of control over personal information is very important in adopting mobile health technologies [57], especially at an older age [58]. Thus, older adults may feel reluctant to use technology that automatically collects data about their every movement [59]. We, therefore, provided a “start/stop walk” button to voluntarily document a walk, similar to the usage policy in several other applications (e.g., the Gramin smartwatch). However, most of the study participants preferred the system to automatically monitor each walk since they tended to forget to start or end a walk on the application. This led to a misunderstanding of the application’s state (whether it was currently monitoring a walk or not), as well as a false representation of performance in the visualization. In addition, the start/stop option caused confusion whenever a participant wanted to take a break while walking. P3 said: “*I didn’t know what’s going on in case I meet a friend or enter a store. Should I end the walk and start another one? The application knows about it?*”. Finally, formally starting each walk emphasized its exercise purpose rather than focusing on the leisure or casual aspects of walking. For example, P8 decided not to document every walk and said: “*I don’t operate the application all the time. When I go to the grocery store, for example, which is a total of 1.4 km, I don’t document the walk*”. P10 felt that there was a negative sense in reporting his actions and related to documentation as “*Turning on the meter*”. It appears that automatic monitoring may be preferable since it entails less interaction with the technology and puts more focus on the activity itself. 

*MyWalk* provided several route suggestions of different lengths for each walk. The purpose was to provide the choice of learning new routes [37] as well as supporting customization of the walk’s length (i.e., suggesting a slightly longer walk than last time), all in the vicinity of the user’s home. However, only 4 out of 12 participants tried the suggested route feature. The rest preferred to do free walks and not commit to a pre-defined route suggestion for several reasons. First, many participants had a preferred, fixed route that they were familiar with. Second, most of the participants were familiar with the routes around their homes and did not feel they wanted or needed walking routes in their known environment. Finally, participants preferred to decide on the difficulty and amount of walking *during* the walk rather than plan it before the walk. For example, P2 said: “*I don’t know how much I will walk and I’m not sure I can commit to a specific distance ahead. I’m not sure how I would feel during the walk.*” To summarize, the users’ strong preferences for spontaneity and control may strengthen the significance of customization over adaptation in such applications.

### 4.3. Learning and Application Adoption

Although the application was designed to be as simple as possible, most participants needed to devote time and effort to learn how to use it (between a few hours to several walks). Based on participant interviews and diaries, we observed a gradual learning process among about half of the participants. This process entailed getting to know the main features and functionalities first (i.e., operating the application while walking), using them for a while, and then continuing to add functionalities when feeling proficient enough.

This resembles an onboarding process targeted at helping users learn new software as well as engaging them in operating it in a guided way [60]. Onboarding is usually used in the setup stage [61] or in the first encounter with new software [59,61,62]. However, learning to adopt a monitoring application that is used sporadically throughout a week entails a longer learning process. We recommend exploring the utility of adopting a long onboarding process, which entails exposing the functionality of the system in a gradual way according to the user’s pace [62]. For example, when opening the application for the first time, instruction can focus on showing the user how to start the walking activity. Only after the activity is completed, and possibly only after several walking sessions, should the application direct users to visualize performance and other functions.

### 4.4. Limitations and Future Work

This study had several limitations. First, only older adults who can walk twice a week for at least 30 min per walk were included in the study. This may have created a selection bias of healthier participants with a positive attitude towards walking. It is also important to explore how to encourage people who do not walk at all to start walking. Second, since we conducted a 2–3-week field study which required careful monitoring of each participant, the sample size was relatively small. Third, most participants stated they felt constrained to comply with the study’s conditions because they agreed to participate. Therefore, statements regarding motivation were addressed very carefully to make sure that they referred to the walk or the application rather than their obligations as study participants. Fourth, due to technical constraints, our system offered circular routes starting from each participant’s home only. This may have caused several of the participants not to use the route suggestions, as they preferred to choose known routes around their homes. Fifth, although wearable devices such as accelerometers are more objective than self-reported questionnaires (e.g., the PASE used in the current study) in assessing participants’ physical activity state, wearables are not yet fully validated for assessing physical activity nor routinely used by older adults [63]. Finally, three of the participants became ill with COVID-19 during the study and therefore their study period was briefly suspended and resumed after they recovered. This break, as well as the illness itself, may have affected their performance and attitude toward walking.

This study raises several future research directions for mobile health applications that support walking in older adults. First, to explore the design space of such products, we used a design probe that monitors the walking activity by adding two features—walking support (i.e., the Home, SOS, and Rest buttons) and route suggestions. Both of these features were seldom used by the participants. The walking support features were perceived as targeted at people with disabilities and not as the safety net we designed them to be. In future work, it will be interesting to explore whether it is possible to provide these options in a more appropriate way. For example, to present them in a positive context rather than focusing on participants’ weaknesses (e.g., the bench button can be presented as *stop and enjoy the fresh air*, the SOS button as *reach out to a friend*, and the home button as *let’s go home*). Moreover, the route suggestions were perceived as restrictive. Based on the types of walks that emerge from our data (see Section 4.1), we assume that planning a walk may be a welcome feature in future technology but further research on how to implement it while keeping users’ sense of control is needed. An interesting suggestion raised by one of the participants was to enable editing of route suggestions by dragging the suggested route line (Figure 3c) on the map. This may provide an interesting option in-between free walks and predefined routes.

Second, with regard to extrinsic motivation during the walk, several interesting suggestions were raised by the participants. Two of the participants suggested providing encouraging notifications (e.g., “Well done!” or “You are doing great, keep on going!”). Another suggestion was to provide contextual information in the form of interesting facts, information, or directions to different points of interest such as historical buildings, or archeological sites while walking. For example, P13 said that: “*We* [his wife and himself] *try to make our walks less technical* [e.g., less about measuring distance or time] *and more interesting*”. Reminders and motivational messages are perceived as supportive and helpful in many health-related behavioral change applications [63,64], specifically those for supporting walking [26,65]. However, they should be designed carefully in order not to decrease intrinsic motivation such as self-competence over time [66]. Providing notifications while walking may also require cognitive resources and may be problematic for older adults [67]. Therefore, providing personalized motivational cues to empower the users in a non-intrusive way is highly important.

Third, as we saw in our study, data visualization is a key element in self-tracking technologies since it enables learning and recognizing trends, thereby supporting behavioral change [66,67,68]. With regard to monitoring technologies for the older population, data visualization was mentioned in previous work as a feature targeted at representing goals [69] and progress [24]. However, there is a lack of guidelines on how to design visualizations for older adults. In our study, we used a simple bar chart visualization; however, more detailed design guidelines on visualization for older adults based on empirical studies should be explored.

## 5. Conclusions

Our goal was to learn more about how to design mobile health walking applications targeted at older adults and to better understand the needs of this population when using such tools. We conducted a field study using a design probe to elicit themes and requirements in this context. Our findings show that the importance of walking is well-known to older adults, but that there are many topics and issues that should be considered when designing a walking application for this population. These include topics related to using the application before the walk (e.g., customization and planning), while walking (e.g., enhancing motivation while walking and walking support), and after walking (e.g., reflecting on performance through visualizations). We expect that the insights gained in this study will be used to design walking applications that are more suitable and better adopted by older adults, and thus, ultimately, enhance active and healthy aging.

## Figures and Tables

**Figure 1 ijerph-20-03611-f001:**
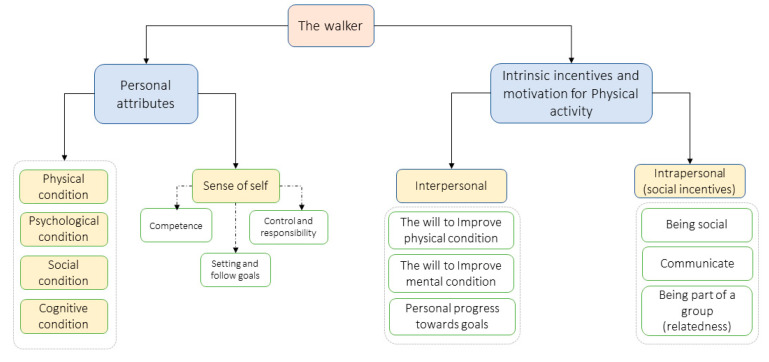
The older walker’s attributes presented using the Personal Investment Theory.

**Figure 2 ijerph-20-03611-f002:**
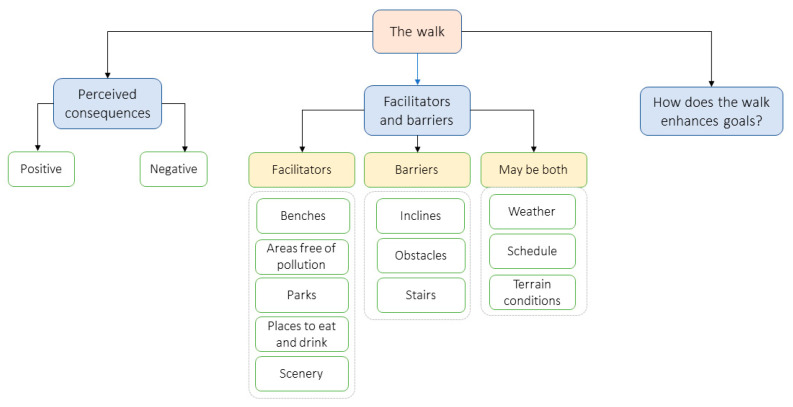
Attributes of the walk presented using the Personal Investment Theory from the older adult walker’s perspective.

**Figure 3 ijerph-20-03611-f003:**
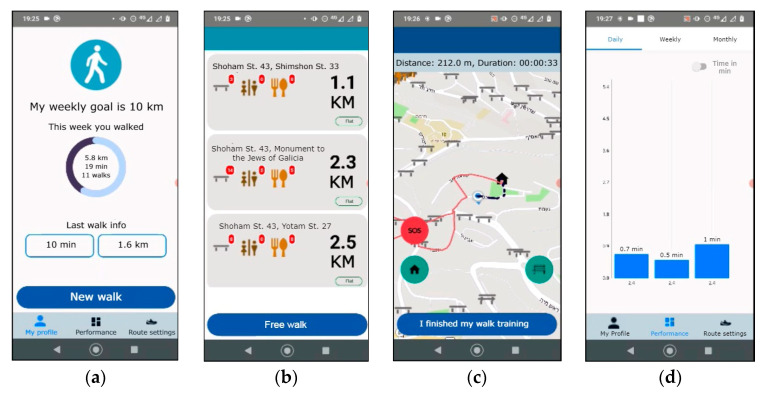
Representative images from MyWalk mobile application: (**a**) main page, (**b**) route suggestions, (**c**) walking session, (**d**) daily duration of walks.

**Table 1 ijerph-20-03611-t001:** Participants’ information.

Part. #	Age	Gender	Chronic Health Condition	Previous Occupation	Daily Dist. (km)	Daily Walk Dur. (min)	Apps in Use	Relevant Measures from Apps
**P1**	69	male	Atrial fibrillation	Logistics in the military	10	120	LG Health	Duration, distance
**P2**	75.5	male	Heart disease		4	45	General pedometer	Steps
**P3**	73	female		Rehabilitation teacher for the visually impaired	1.5	60	General pedometer	Steps
**P4**	75	male	Diabetes, blood pressure, bypass surgery	Business development manager	3.5	90	Google maps and GPS	Location, distance
**P5**	69	female		Microelectronics engineer	2	45	Health insurance app + smartwatch	Steps, distance
**P6**	69	male		Mechanical engineer	2.5	45	None	
**P7**	77.5	male	Inguinal hernia	Independent textile agent	3	25	Health insurance app + smartwatch + General pedometer	Steps, distance
**P8**	70	male		Police investigator	10	210	Altimeter	Altitude
**P9**	70	male		Aeronautical engineer	3	45	Google maps	Progress on the map
**P10**	70.5	male		Administrator in the defense industry	7	60	LG Health	Duration, distance
**P11**	76.5	female	Hip fixation, pelvis fracture	Nurse	4.5	60	General pedometer	Steps
**P12**	69	male		Quality assurance aviation	10	100	none	
**P13**	74	male		Chemist	4.5	60	none	
**P14**	79	male		Engineer	3	40	General pedometer + smartwatch	Steps, distance, duration

**Table 2 ijerph-20-03611-t002:** Participants’ walking details with the application in average minutes and distance per walk.

Part. #	Study Period (Weeks)	# of Walks	Avg Walk Dur. (min)	Avg Walk Dist. (KM)
**P1**	3	16	110.81 (SD = 49.75)	7.21 (SD = 3.36)
**P2**	3	4	78.82 (SD = 33.28)	4.28 (SD = 3.9)
**P3**	3	8	30.24 (SD = 21)	1.3 (SD = 0.87)
**P4**	3	17	78.12 (SD = 29.62)	4.25 (SD = 1.77)
**P5**	3	12	16.39 (SD = 14.54)	0.42 (SD = 0.38)
**P6**	3	7	55.7 (SD = 101.81)	2.93 (SD = 4.33)
**P7**	3	14	33.09 (SD = 23.88)	1.74 (SD = 1.11)
**P8**	2	7	138.51 (SD = 97.75)	8.63 (SD = 4.19)
**P9**	2	8	12.08 (SD = 17.04)	1.18 (SD = 1.26)
**P10**	2	21	17.06 (SD = 11.65)	2.12 (SD = 2.68)
**P11**	2	12	73.48 (SD = 78.25)	3.09 (SD = 2.07)
**P12**	2	12	93.21 (SD = 48.59)	7.07 (SD = 3.69)
**P13**	2	7	43.4 (SD = 9.76)	2.86 (SD = 0.55)
**P14**	2	10	27.51 (SD = 8.03)	0.27 (SD = 0.3)

## Data Availability

Data sharing not applicable.

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
