# Peer review of "Designing Mobile Health Applications to Support Walking for Older Adults"

_ijerph, 2023, doi:10.3390/ijerph20043611_

Round 1
Reviewer 1 Report
I thank the editor for the opportunity to review this manuscript. It's very interesting, and I think it has a lot of potential. However, before deciding to accept it, some important points must be resolved:
1. Please add the trademark symbol when naming devices like Fitbit or strava. Also, put the manufacturer and country in parentheses.
2. In the introduction, it is necessary to add evidence of digital technologies in older adults. In recent years, systematic reviews have appeared showing the usefulness of applications in increasing physical activity and improving function in older adults (Yerrakalva D, et al. Effects of Mobile Health App Interventions on Sedentary Time, Physical Activity, and Fitness in Older Adults: Systematic Review and Meta-Analysis. J Med Internet Res. 2019 Nov 28;21(11):e14343. Solis-Navarro L, et al. Effectiveness of home-based exercise delivered by digital health in older adults: a systematic review and meta-analysis. Age Aging. 2022 Nov 2;51(11):afac243.)
3. Section 2.1.1 explains the physical attributes that determine whether an older adult performs physical activity. However, a very important cause is not discussed, which are chronic pathologies that affect the respiratory system (and generate ventilatory limitation, eg COPD), the cardiovascular system (and can generate vascular involvement, eg pulmonary hypertension or heart disease), musculoskeletal system (which can affect range of motion, eg STROKE), etc. This factor should be added and further incorporated into Figure 1.
4. One factor that determines the level of physical activity is the anthropometric characteristics of the subjects. Will the authors be able to add the body mass index of the participants?
5. The authors must add in limitations that physical activity was evaluated with a questionnaire (PASE) which is not ideal since the best way is through accelerometers (or even double-labelled water).
Reviewer 2 Report
Attached are my comments. The paper is interesting, but there is a bit of a mess in it.
